# Using Machine Learning to Detect Theranostic Biomarkers Predicting Respiratory Treatment Response

**DOI:** 10.3390/life12060775

**Published:** 2022-05-24

**Authors:** Vasilis Nikolaou, Sebastiano Massaro, Masoud Fakhimi, Wolfgang Garn

**Affiliations:** 1Business School, University of Surrey, Guildford GU2 7XH, UK or sebastiano.massaro@theonelab.org (S.M.); masoud.fakhimi@surrey.ac.uk (M.F.); w.garn@surrey.ac.uk (W.G.); 2The Organizational Neuroscience Laboratory, London WC1N 3AX, UK

**Keywords:** biomarkers, machine learning, diagnosis, treatment response

## Abstract

Background: Theranostic approaches—the use of diagnostics for developing targeted therapies—are gaining popularity in the field of precision medicine. They are predominately used in cancer research, whereas there is little evidence of their use in respiratory medicine. This study aims to detect theranostic biomarkers associated with respiratory-treatment responses. This will advance theory and practice on the use of biomarkers in the diagnosis of respiratory diseases and contribute to developing targeted treatments. Methods: We performed a cross-sectional analysis on a sample of 13,102 adults from the UK household longitudinal study ‘Understanding Society’. We used recursive feature selection to identify 16 biomarkers associated with respiratory treatment responses. We then implemented several machine learning algorithms using the identified biomarkers as well as age, sex, body mass index, and lung function to predict treatment response. Results: Our analysis shows that subjects with increased levels of alkaline phosphatase, glycated haemoglobin, high-density lipoprotein cholesterol, c-reactive protein, triglycerides, hemoglobin, and Clauss fibrinogen are more likely to receive respiratory treatments, adjusting for age, sex, body mass index, and lung function. Conclusions: These findings offer a valuable blueprint on why and how the use of biomarkers as diagnostic tools can prove beneficial in guiding treatment management in respiratory diseases.

## 1. Introduction

The use of biological markers (hereafter, biomarkers) as a diagnostic tool to develop targeted treatments—an approach commonly known as theranostics—has been increasingly popular in cancer research. For instance, a recent systematic review [1] suggests that micro RNAs (miRNAs) can be considered theragnostic biomarkers for predicting radiotherapy response. Another review [2] found that miRNAs can also be used as biomarkers in the diagnosis of prostate cancer, and it can potentially have an impact on chemotherapy response. Moreover, Jothimani et al. [3] explored the role of non-coding RNAs (ncRNAs) as diagnostic biomarkers and therapeutic agents for colorectal cancer. Nair et al. [4] suggest that neutrophil gelatinase-associated lipocalin (NGAL) can be considered as a diagnostic biomarker for perihilar cholangiocarcinoma (PHC) and potentially for developing targeted therapeutics, while Tung et al. [5] supported that using miRNAs as a diagnostic biomarker for colon adenocarcinoma (COAD) has led to a potential targeted drug (Gemcitabine) for this type of gastrointestinal cancer.

The role of theranostic biomarkers is not limited to cancer, and it has also been explored in other chronic diseases, including Alzheimer’s disease (AD) and hepatitis, among others. Thus, Mahaman et al. [6] reviewed several biomarkers that can be used for the accurate and the early diagnosis of AD, yielding in the development of targeted treatments. Portelius et al. [7] investigated the performance of truncated amyloid-β (Aβ) isoforms as theragnostic markers for AD, and they supported their use for developing potential treatments. Atkinson et al. [8] explored the association between serum keratin-18 (K18) and histological features in patients with severe alcoholic hepatitis (AH). They found a strong association, suggesting that serum K18 levels can be used as a theranostic biomarker in the early diagnosis and appropriate treatment of AH.

As for the role of theranostic biomarkers in respiratory medicine, there is limited research. Among this, we found convincing evidence for the use of blood eosinophils. For instance, Kerkhof et al. [9] found that elevated blood eosinophils associated with increased exacerbations in patients with mild-to-moderate Chronic Obstructive Pulmonary Disease (COPD) can lead to improved lung function when treated with inhaled corticosteroids (ICS). The theranostic ability of blood eosinophils was also confirmed by Siddiqui et al. [10], who found that higher blood eosinophils count in patients with COPD was associated—when treated with ICS—with decreased exacerbations as well from a pooled analysis of ten studies with a total of 85,059 patients with COPD [11]. The latter study confirmed the association between blood eosinophil count and reduced (or increased) exacerbations by escalating (or de-escalating) ICS in patients with COPD.

This study aims to build on this evidence toward expanding the search for theranostic biomarkers associated with respiratory-treatment response. Specifically, rather than focusing on previously known and validated biomarkers (e.g., blood eosinophils), we will identify a novel set of theranostic biomarkers whose response to any respiratory treatment has not yet fully been explored. We will do this in a large sample of healthy individuals, a small proportion of whom are exposed to respiratory treatment. This will help us gain an advanced understanding of the role of biomarkers in disease diagnosis towards drug development.

## 2. Materials and Methods

This is a retrospective study on adults who undertook the UK Household Longitudinal survey “Understanding Society” [12]. In this survey, information is collected annually on household changes and individual circumstances. During the period 2010–2012, all adults aged 16 and over were invited to participate in a nurse health assessment interview that consisted of a range of physical measures and biomarkers.

From a total of 35,937 participants eligible for the nurse visit, 20,700 participated in the health assessment. Of those, 14,333 participants agreed to give their blood sample for biomarker analysis and 13,102 participants (36.5%) had at least one biomarker available (Figure 1).

Missing values for biomarkers (ranging from 2% to 40%), weight (3%), height (1%), percent predicted forced expiratory volume in 1 s (ppFEV1) (36%), and body mass index (BMI) (3%) were imputed with the method of multivariate imputation of chained equations [13]. We then standardized the biomarkers to the same scale and performed recurrent feature extraction (i.e., backwards feature selection) [14], an unbiased and data-driven method that takes into account all available biomarkers to identify those significantly associated with any respiratory treatment response. We used the repeated cross-validation resampling method with ten repeated training/test splits of the data during feature elimination to mitigate overfitting [15]. The biomarkers derived from the recurrent feature extraction along with age, sex, ppFEV1, and BMI were used as predictors for training several machine learning models (logistic regression, decision tree [16], random forest [17], and gradient boosting machine [18]) on a 70% random split of the data. The remaining 30% of the data was used for validation. To ensure the continuous predictors (i.e., biomarkers, age, ppFEV1, and BMI) were on the same scale, we standardized them prior to training (and testing). To deal with class imbalance, due to the small number of participants receiving respiratory treatment, we used the R package “ROSE” for random over-sampling of the minority class [19]. The models’ performance was assessed on overall accuracy, sensitivity, specificity, positive predictive values (PPV), and negative predictive values (NPV) [20]. We used a logistic regression model on the whole dataset to interpret the association of biomarkers on treatment response after adjusting for age, sex, ppFEV1, and BMI. Adjusted odds ratios and 95% confidence intervals (CIs) were used to assess the impact of biomarkers on respiratory drug use.

## 3. Results

The demographic characteristics of our sample are described in Table 1.

As shown in Table 1, the study’s participants have an average age of 52 years, most of them are female with an average BMI of 28 corresponding to an overweight category [21].

Table 2 summarizes the participants’ clinical characteristics including biomarkers, percent predicted lung function, and respiratory drug use.

As shown in Table 2, all biomarkers and the lung function (ppFEV1) fall within a normal range, while a small proportion of participants had received any respiratory drug, suggesting that this is a healthy group of people. Following missing values imputation and recurrent feature extraction, we retrieved 16 biomarkers significantly associated with respiratory drug use. These are: albumin, alkaline phosphatase, aspartate transaminase, cholesterol, dehydroepiandrosterone sulphate, gamma-glutamyltransferase, glycated hemoglobin, high-density lipoprotein, c-reactive protein, insulin-like growth factor 1, ferritin, triglycerides, urea, haemoglobin, fibrinogen activity (Clauss), and cytomegalovirus (cmv) IgG.

We trained four machine learning models (i.e., logistic regression, decision tree, random forest, and gradient boosting machine) in the training dataset, and we assessed their performance in predicting treatment response on the validation set (Table 3).

The logistic regression model performs better with response to sensitivity, i.e., the ability to correctly predict treatment response. In contrast, the other three models (i.e., random forest, gradient boosting machine, and decision tree) perform better than the logistic regression in terms of specificity, which is the ability to correctly rule out any respiratory drug use. All models have a similar performance in PPV or NPV, i.e., in correctly predicting (or ruling out) treatment response given that the participant was receiving (or not) any respiratory therapy.

As the logistic regression model exhibited a higher sensitivity compared to other models (64% vs. 54%), we used this to interpret the impact of biomarkers on treatment response after adjusting for age, sex, body mass index, and lung function (i.e., ppFEV1) (Table 4).

As shown in Table 4, on the one hand, increased levels of half of these biomarkers (i.e., alkaline phosphatase, glycated haemoglobin, high-density lipoprotein cholesterol, c-reactive protein, triglycerides, haemoglobin, and fibrinogen) were significantly associated with higher odds of treatment response. On the other hand, participants with increased levels of albumin, cholesterol, dehydroepiandrosterone sulphate, insulin-like growth factor 1, and ferritin were significantly less likely to receive any respiratory treatment. The associations between aspartate transaminase, gamma glutamyl transferase, urea, cytomegalovirus IgG and respiratory treatment use were not statistically significant.

## 4. Discussion

This study used recurrent feature extraction—a data reduction method—to identify the most significant biomarkers associated with respiratory treatment response. We trained several machine learning models on 70% of the data, and we validated their performance on 30% of the data to identify that a logistic regression model was the most sensitive (64%) to treatment response. Among the biomarkers associated with increased odds of respiratory treatment, five of them (high-density lipoprotein cholesterol, c-reactive protein, triglycerides, haemoglobin, and Clauss fibrinogen) are risk factors for cardiovascular disease (CVD), and two of them (glycated haemoglobin and alkaline phosphatase) are risk factors for diabetes and liver disease, respectively. The link between liver disease and CVD and diabetes and CVD has been confirmed in previous studies [22,23]. Recent studies have also demonstrated the association between CVD, liver disease, diabetes, and respiratory viral infections (e.g., COPD and COVID-19) [24,25,26]. Therefore, our findings are consistent with previous studies suggesting that people at risk of CVD, liver disease, or diabetes are also at risk of respiratory infections and therefore can be treated similarly.

In contrast, increased cholesterol and albumin levels as well increased levels of ferritin and insulin-like growth factor 1 were associated with reduced odds of respiratory treatment, which may suggest that either a) these participants were more likely to receive other than respiratory treatments—in fact, the proportion of participants who received any cardiovascular drug was higher than that of any respiratory treatment (28% vs. 12%)—or b) it was a result of negative confounding [27] that led to an underestimation of a true association, which is frequently seen in observational studies [28].

There are some limitations to our study. First, the absence of eosinophil counts, which is proven to be associated with respiratory diseases, e.g., COPD and mediated by inhaled corticosteroids [9,10,11] may be the reason for the low accuracy—especially the sensitivity—of our models. Although this study aims to explore the association of other than eosinophils biomarkers, we believe that had this biomarker been present in the dataset, our models would have been more sensitive in predicting any respiratory treatment response. Another limitation is that our study consists of a sample of healthy participants, whose measured biomarkers are within a normal range and, consequently, a small proportion of them are receiving respiratory-related treatment. Therefore, despite the biomarkers included in this study being measures of risk factors for potential CVD, liver disease, or diabetes, we would not be able to identify any direct link between those biomarkers and any disease that would potentially lead to respiratory treatment.

Moreover, approximately 5000 participants did not consent or were unable to give their blood, so their biomarkers could not be assessed. These participants could be different from those who have given blood samples, and their inclusion in the study may have altered the results. A fourth limitation is the lack of specific respiratory related treatments in our data. Our treatment response consists of any respiratory drug taken without being specific on the kind of drug, its dosage, or the frequency taken. Therefore, we could not assess whether any of the identified biomarkers were associated with a particular drug that would help provide a roadmap for targeted treatments.

## 5. Conclusions

This is the first study that, to the best of our knowledge, presents a set of biomarkers—known to be associated with chronic diseases (e.g., CVD, liver disease, and diabetes)—whose association with a respiratory treatment response has not been previously explored. This study was done on a large sample of healthy participants with unknown underlying conditions and a low intake of respiratory treatment. We used machine learning and data reduction methods to identify the most significantly associated biomarkers with any respiratory treatment response.

We trained several machine learning models, including logistic regression and random forest, with 70% of the data, and we assessed their performance on the rest of the data (30%) that did not contribute any information to the models’ development (i.e., they were independent). Although none of these biomarkers are a known risk factor for respiratory disease, we identified 16 of them, and along with age, sex, body mass index, and lung function we were able to predict respiratory treatment response with 64% accuracy correctly. We then used the logistic regression model to calculate the odds of association between the biomarkers and the treatment response. We found that elevated levels of alkaline phosphatase, glycated haemoglobin, high-density lipoprotein cholesterol, c-reactive protein, triglycerides, haemoglobin, and fibrinogen activity were associated with increased odds of treatment response, whereas increased levels of albumin, cholesterol, dehydroepiandrosterone sulphate, insulin-like growth factor 1, and ferritin were associated with reduced odds of respiratory treatment response. There was insufficient evidence for a significant association between aspartate transaminase, gamma glutamyl transferase, urea, cytomegalovirus IgG, and respiratory treatment response.

Should our findings be validated in other populations—including, e.g., patients with respiratory diseases with a variety of respiratory drugs—we are confident that they would assist in effectively guiding both disease diagnosis and associated targeted therapies.

## Figures and Tables

**Figure 1 life-12-00775-f001:**
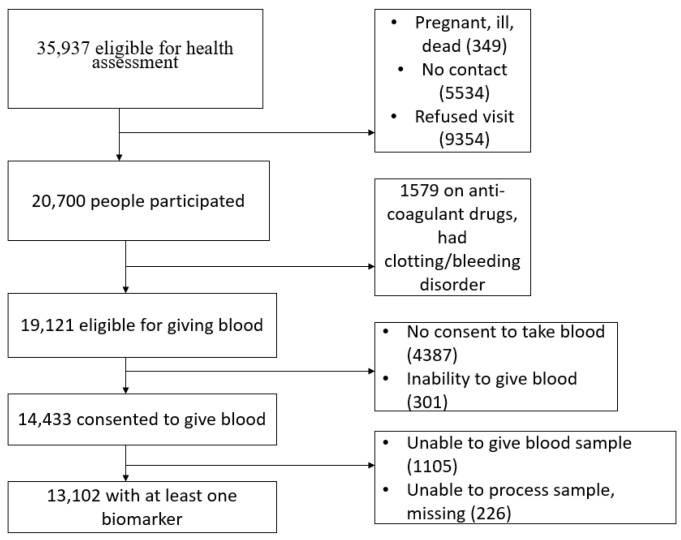
Reduction flow chart of participants for the study.

**Table 1 life-12-00775-t001:** Participants’ demographic characteristics.

Characteristic	Statistic	Participants (N = 13,102)
Age (years)	n	13,102
	Mean (SD)	51.5 (17.2)
	Median	52.0
Sex, n (%)	Male	5848 (45)
	Female	7254 (55)
Height (cm)	n	12,988
	Mean (SD)	167.5 (9.5)
	Median	167
Weight (kg)	n	12,773
	Mean (SD)	78.2 (16.1)
	Median	76.6
BMI (kg/m^2^)	n	12,749
	Mean (SD)	27.8 (5.3)
	Median	27.2

BMI: Body Mass Index; SD: Standard Deviation.

**Table 2 life-12-00775-t002:** Participants’ clinical characteristics.

Characteristic	Statistic	Participants (N = 13,102)
Cholesterol (mmol/L)	n	12,895
	Mean (SD)	5.4 (1.2)
	Median	5.3
HDL cholesterol (mmol/L)	n	12,876
	Mean (SD)	1.6 (0.5)
	Median	1.5
Triglycerides (mmol/L)	n	12,898
	Mean (SD)	1.8 (1.2)
	Median	1.5
Glycated haemoglobin (mmol/mol)	n	12,162
	Mean (SD)	37.3 (8.2)
	Median	36
C-reactive protein (mg/L)	n	12,530
	Mean (SD)	3.3 (7.1)
	Median	1.4
Cytomegalovirus IgG (cmv)	n	12,896
	Mean (SD)	1.5 (0.5)
	Median	2
Cytomegalovirus IgM (cmv)	n	12,896
	Mean (SD)	1.9 (0.2)
	Median	2
Clauss fibrinogen (g/L)	n	12,837
	Mean (SD)	2.8 (0.6)
	Median	2.7
Haemoglobin (g/L)	n	12,156
	Mean (SD)	136.9 (13.9)
	Median	137
Ferritin (ug/L)	n	12,894
	Mean (SD)	137.4 (176.8)
	Median	100
Albumin (g/L)	n	12,920
	Mean (SD)	46.8 (2.9)
	Median	47
Alkaline phosphatase(u/L)	n	12,785
	Mean (SD)	71.5 (23.4)
	Median	69
Alanine transaminase(u/L)	n	12,777
	Mean (SD)	28 (26.2)
	Median	23
Aspartate transaminase(u/L)	n	12,386
	Mean (SD)	30.7 (23.6)
	Median	29
Gamma glutamyl transferase (u/L)	n	12,816
	Mean (SD)	34.1 (51.3)
	Median	23
Creatinine (µmol/L)	n	12,918
	Mean (SD)	76.4 (19.6)
	Median	74
Urea (mmol/L)	n	12,923
	Mean (SD)	6.2 (1.7)
	Median	6
Testosterone (nmol/L)	n	7830
	Mean (SD)	11.8 (8.0)
	Median	12.6
Insulin-like growth factor 1 (nmol/L)	n	12,831
	Mean (SD)	18.4 (7.4)
	Median	17
Dehydroepiandrosterone sulphate (µmol/L)	n	12,873
	Mean (SD)	4.6 (3.2)
	Median	3.8
ppFEV1	n	8471
	Mean (SD)	92.9 (16.4)
	Median	94.1
Respiratory drug use, n (%)	Yes	1526 (12)
	No	11,576 (88)

HDL: high-density lipoprotein; SD: Standard Deviation; ppFEV1: percent predicted Force Expiratory Volume in 1 s.

**Table 3 life-12-00775-t003:** Models’ performance on the validation dataset.

Logistic Regression	Observed	
Predicted	No treated	Treated
No treated	2130	162
Treated	1342	295
Accuracy (%)	62	
Sensitivity (%)	64	
Specificity (%)	61	
PPV (%)	18	
NPV (%)	93	
**Decision Tree**		
**Predicted**		
No treated	2293	208
Treated	1179	249
Accuracy (%)	65	
Sensitivity (%)	54	
Specificity (%)	66	
PPV (%)	17	
NPV (%)	92	
**Random Forest**		
**Predicted**		
No treated	2475	208
Treated	997	249
Accuracy (%)	69	
Sensitivity (%)	54	
Specificity (%)	71	
PPV (%)	20	
NPV (%)	92	
**Gradient Boosting machine**		
**Predicted**		
No treated	2462	208
Treated	1010	249
Accuracy (%)	69	
Sensitivity (%)	54	
Specificity (%)	70	
PPV (%)	20	
NPV (%)	92	

PPV: Positive predicted value; NPV: Negative predicted value.

**Table 4 life-12-00775-t004:** Impact of biomarkers on treatment response.

Biomarker	Odds Ratio	95% CI	*p*-Value
Albumin (g/L)	0.94	0.91, 0.98	<0.001
Alkaline phosphatase(u/L)	1.03	1.007, 1.07	0.017
Aspartate transaminase(u/L)	1.03	0.99, 1.06	0.106
Cholesterol (mmol/L)	0.94	0.91, 0.97	<0.001
Dehydroepiandrosterone sulphate (µmol/L)	0.78	0.75, 0.81	<0.001
Gamma glutamyl transferase (u/L)	1.03	0.99, 1.06	0.081
Glycated haemoglobin (mmol/mol)	1.05	1.02, 1.08	0.001
HDL cholesterol (mmol/L)	1.08	1.04, 1.11	<0.001
C-reactive protein (mg/L)	1.04	1.003, 1.07	0.035
Insulin-like growth factor 1 (nmol/L)	0.92	0.88, 0.95	<0.001
Ferritin (ug/L)	0.91	0.87, 0.94	<0.001
Triglycerides (mmol/L)	1.06	1.02, 1.09	<0.001
Urea (mmol/L)	0.99	0.96, 1.02	0.467
Haemoglobin (g/L)	1.04	1.01, 1.08	0.009
Fibrinogen activity (Clauss) (g/L)	1.09	1.06, 1.13	<0.001
Cytomegalovirus IgG (cmv)	0.97	0.94, 1.002	0.069

CI: Confidence interval; HDL: high-density lipoprotein. Adjusting for age, sex, body mass index, and ppFEV1.

## Data Availability

Access to the data can be found here: The UK Household longitudinal Study (https://www.understandingsociety.ac.uk/ (accessed on 31 July 2021)).

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
