# Peer review of "Using Machine Learning to Detect Theranostic Biomarkers Predicting Respiratory Treatment Response"

_life, 2022, doi:10.3390/life12060775_

Round 1
Reviewer 1 Report
This article is about theranostics-the use of diagnostics for developing targeted therapies. This study aims to detect theranostic biomarkers associated with respiratory-treatment responses.
A sample 18 of 13,102 adults from the UK household was analyzed and 16 biomarkers were identified which are associated with respiratory treatment responses.
The article is written in clear and concise language and it easy to read.
Please, edit the style of citations
Author Response
Thanks for your comment. We follow the journal’s recommended ACS style for the references, i.e., 1. Author 1, Author 2. Title of the article. Journal. Year, page numbers. We understand that some references however have a different font, and this now has been edited (references: 5, 7, 9, 12, 13, 15 - 28). All in all, we have addressed your comments and would like to thank you once more for your positive feedback on our work.
Reviewer 2 Report
- The title is interesting and the authors works hard on it but still there are a number of things that should be modify.
- The authors are advised to update the abstract section and add more information with respect to your title.
- The introduction part of the manuscript is not written well which constitutes the lack of crux in it. This section is more orientated to disease like cancer and AD rather than the concern topic.
- In abstract section the authors have mentioned that they performed a cross-sectional analysis whereas in the material and method it is retrospective study. Please make it uniform throughout the manuscript.
- The authors are advised to clearly mention the rationale behind feature selection of 16 biomarkers.
- Please do add in manuscript which biomarkers specify squamous or non-squamous cell lung carcinoma.
- Since data collection period was 2010-2012, however study is submitted now. Please justify?
- At line 58 please mention the full form of COPD first. Later on, you can use this abbreviation. Kindly check this matter throughout the manuscript.
- At line 60 Siddiqui et Al. is there please make et al in small letter.
- The manuscript needs standard review.
- Kindly add the future prospective of this study
Author Response
The title is interesting and the authors works hard on it but still there are a number of things that should be modify. Thanks for your appreciation of our work. As elaborated in the responses below we worked hard to tackle all the issues that you have raised in your report.
- The authors are advised to update the abstract section and add more information with respect to your title. Thank you for your comments. Unfortunately, due to the 200-words limit, we are strongly bounded in relation to the information that we could provide in the abstract. Nonetheless, we followed your advice and added a couple of informative, new sentences with respect to machine learning models (lines 21-22).
- The introduction part of the manuscript is not written well which constitutes the lack of crux in it. This section is more orientated to disease like cancer and AD rather than the concern topic. Thanks for your comment. We are sorry you feel that way. Please allow us to clarify why we included information on cancer and AD to begin with. Following the journal’s instructions, authors are tasked to include a review of the current state of the research field. Because our research focuses on theranostic biomarkers (vs. respiratory diseases per se, which is the application setting), we have extensively reviewed the current literature and reported that such biomarkers are predominately used in cancer research (lines 33-43). Similarly, another popular field for using theranostic biomarkers is Alzheimer’s disease (lines 45-50) and alcoholic hepatitis (lines 51-54). Following this briefing, we respectfully note that we diligently highlighted the currently limited research on theranostics in respiratory medicine and reported the role of eosinophils as – to the best of our knowledge - the only known theranostic biomarker investigated thus far (lines 56-66). To reiterate our focal concern on respiratory issues we stress the main aim of our study (lines 68-72) and its future perspectives (lines 74-75). We believe that our actions make our introduction comprehensive and in line with the journal’s editorial requests.
- In abstract section the authors have mentioned that they performed a cross-sectional analysis whereas in the material and method it is retrospective study. Please make it uniform throughout the manuscript. Thanks for this comment. We refer to “cross-sectional analysis” to indicate that the statistical analysis was performed at one time point, i.,e., cross-sectionally. The study itself, however is a retrospective study, as it was carried out with previously collected data. To avoid any possible confusion, we would like to underscore that we relate the term cross-sectional to the analysis rather than to the study itself and this use is uniform across the manuscript.
- The authors are advised to clearly mention the rationale behind feature selection of 16 biomarkers. Thanks for the comment. We added a sentence (lines 97-98) to explain that recurrent feature extraction is an unbiased and data-driven method to identify those biomarkers that are significantly associated with respiratory treatment response. We would like to stress that the number of biomarkers was not known in advance. It was derived from the analysis and reported in the results section (lines 134-140)
- Please do add in manuscript which biomarkers specify squamous or non-squamous cell lung carcinoma. We thank you for this comment that however left us slightly puzzled as we failed to appreciate its link to our study. We investigate the association of biomarkers and respiratory treatment response, not of cancer. We would be happy to take further actions may you please clarify the rationale for this request in response to the treatment response which is the focus of our study (vs treatment strategy or kind of disease per se). Thanks.
- Since data collection period was 2010-2012, however study is submitted now. Please justify? Thanks for your comment. The UK Household Longitudinal survey started in 2009 and is ongoing (https://www.understandingsociety.ac.uk/documentation/mainstage/technical-reports). During its second wave (2010-2012), biomarker data were recorded for 13102 participants. In our study, we used this sample to explore the association of biomarkers and respiratory treatment response. As we underline in the conclusions section, such association had not been previously explored (lines 218-220). These are secondary available datasets currently widely used in many publications across fields.
- At line 58 please mention the full form of COPD first. Later on, you can use this abbreviation. Thanks for the comment. It is now updated accordingly (line 59). Kindly check this matter throughout the manuscript. We checked other abbreviations and acronyms and can confirm they are all written in their full form first.
- At line 60 Siddiqui et Al. is there please make et al in small letter. Thanks for the comment. The issue is now resolved (line 61).
- The manuscript needs standard review. We are unsure about the actions you would like us to take in relation to this comment, however we can confirm that all authors have reviewed and approved the manuscript.
- Kindly add the future prospective of this study Thanks for your comment. This is now mentioned in the introductory section (lines 74-75) and in the conclusions section (lines 240-242).